# Real-Life Comparison of Fosfomycin to Nitrofurantoin for the Treatment of Uncomplicated Lower Urinary Tract Infection in Women

**DOI:** 10.3390/biomedicines11041019

**Published:** 2023-03-27

**Authors:** Asher Shafrir, Yonatan Oster, Michal Shauly-Aharonov, Jacob Strahilevitz

**Affiliations:** 1Faculty of Medicine, Hebrew University of Jerusalem, Jerusalem 9190401, Israel; 2Department of Gastroenterology, Hadassah-Hebrew University Medical Center, Jerusalem 9112001, Israel; 3Meuhedet Health Services (HMO), Jerusalem 9780207, Israel; 4Department of Clinical Microbiology and Infectious Diseases, Hadassah Hebrew University Medical Center, Jerusalem 9112001, Israel; 5The Jerusalem College of Technology, Jerusalem 9116001, Israel

**Keywords:** urinary tract infections, real life, fosfomycin, nitrofurantoin, adherence

## Abstract

In this study, we compared the failure rates of fosfomycin and nitrofurantoin for uncomplicated urinary tract infections. We used Meuhedet Health Services’ large database to collect data on all female patients, older than 18 years, who were prescribed either antibiotic during 2013–2018. Treatment failure was a composite endpoint of hospitalization, emergency-room visit, IV antibiotic treatment, or prescription of a different antibiotic, within seven days of the initial prescription. Reinfection was considered when one of these endpoints appeared 8–30 days following the initial prescription. We found 33,759 eligible patients. Treatment failure was more common in the fosfomycin group than the nitrofurantoin group (8.16% vs. 6.87%, *p*-value < 0.0001). However, reinfection rates were higher among patients who received nitrofurantoin (9.21% vs. 7.76%, *p*-value < 0.001). Among patients younger than 40 years, patients treated with nitrofurantoin had more reinfections (8.68% vs. 7.47%, *p* value = 0.024). Treatment failure rates were mildly higher in patients treated with fosfomycin, despite having less reinfections. We suggest that this effect is related to a shorter duration of treatment (one vs. five days) and encourage clinicians to be more patient before declaring fosfomycin failure and prescribing another antibiotic.

## 1. Introduction

Urinary tract infections (UTI) are one of the most common infections in primary care settings [1], where empiric antibiotics are prescribed. Current guidelines allow several oral antibiotics with sufficient bioavailability and good adverse reaction profile [2,3], including nitrofurantoin, trimetoprim–sulfamethoxazole, fosfomycin, or pivmecillinam, all to be given similar grades (A-I), i.e., are strongly recommended as first-line treatments, with good supporting evidence.

Several comparisons between the clinical outcomes of urinary infections with these antibiotics were conducted, without a clear advantage for a specific one [4,5,6,7]. The two most recommended antibiotics are nitrofurantoin and fosfomycin. Both antimicrobials share a good general safety profile, adequate concentration in urine, a narrow spectrum, and are being used in many primary care settings. Fosfomycin is the drug with the shortest duration of treatment, typically given as a single 3000 mg dose. This feature of fosfomycin makes it an attractive option, overcoming adherence issues [4]. Although Nitrofurantoin needs to be administered thrice daily for five to seven days, it was found to have a slightly better efficacy compared with fosfomycin [2]. Because each antimicrobial has its merits, it would be beneficial to study what additional factors affect a better patient–drug matching.

The aim of this trial was to evaluate the factors affecting failure among patients with UTI attending primary care in a large group of patients in real-life settings. We used data extracted from electronic medical records (EMRs) of Meuhedet Health Services (MHS), a large Israeli Health Maintenance Organization (HMO), to assess the frequency of different outcomes that constitute treatment failure, according to the antibiotic regimen that was given by primary care physicians. The “off-label” practice of prescribing several consecutive doses of fosfomycin in urinary infection is not allowed in the electronic prescribing forms used by Meuhedet Health Services and therefore was not studied.

## 2. Materials and Methods

Meuhedet is the third largest healthcare provider in Israel, serving over 1,200,000 patients that are representative of the general population of Israel. Meuhedet’s computerized database includes real-time input from all physician visits, diagnoses, medical prescriptions and dispensing data, laboratory results and hospitalizations.

In this retrospective study, we collected data of all female patients aged 18 or above who were diagnosed with urinary tract infection (UTI), simple cystitis (ICD-9 codes—599, 595.5, 595, 595.4, 595.8, 595.9, 595.89) and were prescribed either nitrofurantoin or fosfomycin between January 2013 and December 2018. Data included the patient’s age, serum creatinine level—if measured 90 days (or less) prior to UTI diagnosis—and the following information regarding the 30 days after diagnosis: emergency room (ER) visits, admissions to hospital, visits in a community emergency clinic, intravenous (IV) treatment (gentamicin or ceftriaxone), diagnosis of pyelonephritis, and changes in antibiotic treatment.

Patients were excluded if they had one or more of the following: a positive HCG test in the 270 days prior to UTI diagnosis; a diagnosis of UTI less than 90 days prior to the current diagnosis; diagnosis of pneumonia, upper respiratory tract infection, cellulitis, or fever of unknown origin within the 30 days following the UTI diagnosis.

Treatment was considered a failure if one or more of the following events (ordered from the most severe to the least severe) occurred within seven days from the index UTI diagnosis: hospitalization, emergency room or emergency clinic visit, and IV antibiotic treatment or prescription of a different antibiotic than the initial one. As a change in antibiotic treatment might reflect reasons other than treatment failure, such as an allergic reaction, a change in working diagnosis, and issues with pharmacy supply, we performed an additional analysis of treatment failure while disregarding this endpoint.

Similarly, a patient was considered as having a reinfection if one or more of the following events (ordered from the most to the least severe) happened within eight to thirty days from the initial diagnosis: hospitalization, emergency room or emergency clinic visit, another visit at the primary care physician with a diagnosis of UTI, or a prescription of any antibiotic treatment.

In addition, a patient who had undergone more than one of the events that was defined as a treatment failure was counted as one failure, classified according to the most severe event. The same classification was performed for re-infected patients. Patients defined as treatment failure could not be defined again as re-infected.

### Statistical Analysis

Statistical analysis was performed using R software (R Development Core Team, 2018, R Foundation for Statistical Computing, Vienna, Austria.). The packages ‘ggplot2′, ‘dplyr’, ‘zoo’, ‘data.table’, and ‘tidy’ were used in addition to the default R packages. Categorical variables were summarized as counts and percentages. Continuous variables were summarized as means and standard deviations (SD). Univariate analysis was performed using chi-squared test to compare categorical variables, and Welch’s t-test to compare means of continuous variables. *p*-value of less than 0.05 was considered statistically significant. In addition to *p*-values, the absolute effect size (i.e., the magnitude of the difference between fosfomycin and nitrofurantoin) was calculated; the effect size of two proportions was defined according to Cohen’s *h* [8], which quantifies the size of the difference, allowing us to decide if the difference was meaningful. Number needed to treat (NNT) were also computed to communicate the effectiveness of fosfomycin vs. nitrofurantoin. Multivariate logistic regression was performed to assess the effect of antibiotic treatment on treatment failure and reinfection while controlling for patient characteristics that had been found to be potential confounders; variables with a *p*-value of less than 0.2 in a univariate analysis were considered as potential confounders (and thus were included in the covariate selection process in the multivariate analysis). These were time from last UTI event, age, and creatinine levels.

The study was approved by Meuhedet internal ethics board on 17 February 2020, approval code 01-29-01-20.

## 3. Results

Between 1 January 2013 and 31 December 2018, there were 569,897 visits of women aged 18 and above with a diagnosis of UTI at Meuhedet. Cases were excluded if one or more of the following occurred: (1) no antibiotic treatment was prescribed, (2) the patient had a sequential diagnosis of upper respiratory tract disease, fever of unknown origin, cellulitis, or pneumonia within 30 days from original diagnosis, (3) the patient had a positive HCG test in the past 270 days, or (4) when there was a previous case of UTI 90 days prior to the index event. In addition, patients that were treated with an antibiotic different than fosfomycin or nitrofurantoin were excluded. After these exclusions, 33,759 patients remained who were treated with nitrofurantoin or fosfomycin (Figure 1).

Patients who were treated with nitrofurantoin were significantly older than patients who were treated with fosfomycin (50.6 years vs. 45.2 years, respectively, *p*-value < 0.001) and their previous documented diagnosis of UTI (by the same UTI definitions) was closer in time (407 days vs. 506 days, *p*-value < 0.001) (Table 1). BMI and creatinine levels were mildly higher among patients receiving nitrofurantoin.

Treatment failure was more common among those who were treated with fosfomycin than among those who were treated with nitrofurantoin (8.16% vs. 6.87%, *p* value < 0.0001). However, reinfection rates were higher among patients who received nitrofurantoin (9.21% vs. 7.76%, *p*-value < 0.001) (Table 1, Appendix A). The magnitude of differences between fosfomycin vs. nitrofurantoin was small in all outcomes. The (absolute) NNT here is the average number of patients that need to be treated with the antibiotic medication for one of them to benefit from the medication, compared with the other antibiotic. All NNTs were high, thus showing only small differences between the two treatment groups. 

In a multivariate logistic regression, controlling for age, time from last UTI diagnosis and creatinine, and the use of nitrofurantoin were associated with significantly less treatment failures (odds ratio 0.74 95% CI 0.59–0.92; *p*-value = 0.008, beta = 0.7377, McFadden R^2^ = 0.85) (Table 2).

When excluding a change of antibiotic treatment from the criteria of failure events, a similar multivariate logistic regression controlling the same variables showed that nitrofurantoin was associated with more treatment failures (odds ratio 1.41 95% CI 1.01–1.98; *p*-value = 0.044, beta = 1.4147, McFadden R^2^ = 0.857).

### Subgroup Analysis

Among patients younger than 40, nitrofurantoin and fosfomycin had similar rates of treatment failure (7.96% in both groups, *p* value = 1). In this age group, patients treated with nitrofurantoin had more cases of reinfection (8.68% vs. 7.47%, *p* value = 0.024), derived mainly from a higher likelihood of turning to an ER, and were admitted within seven days of diagnosis. They were, however, less likely to have a change in antibiotic treatment (Table 3, Appendix A). A multivariate logistic regression controlling for age, creatinine, and time from last UTI, showed that nitrofurantoin and fosfomycin had similar outcomes (*p*-value = 0.56).

Among patients 40 years and older, treatment failure was more common with fosfomycin (8.31% vs. 6.25% *p*-value < 0.001), but reinfection was more common with nitrofurantoin (9.51% vs. 7.98%, *p*-value = 0.0004) (Table 3, Appendix A). In a multivariate logistic regression analysis controlling for creatinine, age, and time from last UTI diagnosis, nitrofurantoin treatment failed less than fosfomycin (*p*-value = 0.001; McFacdden R^2^ = 0.796 OR 0.67 95% CI 0.52–0.85). Regarding reinfections, however, in a multivariate logistic regression controlling for age, creatinine, and time from last UTI diagnosis, there was no significant difference between the two treatments (*p*-value = 0.398, McFadden R^2^ = 0.76).

## 4. Discussion

Prescribing antibiotics to women presenting uncomplicated UTI to primary care clinics is not as simple a decision as it seems. Despite the known antimicrobial activity of the many options, additional considerations must be made—the predicted adherence of the specific patient to the suggested drug being one of the most important ones.

In a large multinational, open-labeled, analyst-blinded, randomized clinical trial study published in 2018, Huttner et al. [9] had compared the efficacy of nitrofurantoin and fosfomycin in women with uncomplicated cystitis, and found that five-day nitrofurantoin had significantly better outcomes compared to a single dose of fosfomycin (70% vs. 58% clinical resolution, 74% vs. 63%, 28-day microbiological resolution). Nevertheless, that trial recruited patients hospitalized for other reasons, as well as walk-in clinics in large hospitals, and we suspect that this population does not accurately represent the primary care of acute uncomplicated infections. In another real-life study of 143 patients, there was no statistically significant difference between nitrofurantoin and fosfomycin [10].

Our real-life retrospective study included 33,759 female patients with uncomplicated UTI in the community settings and compared the clinically relevant outcomes of two common antibiotics, fosfomycin and nitrofurantoin, from 1 January 2013 to 31 December 2018. Despite the 4 years that passed since the conclusion of the data collection, these two antibiotics are still the mainstay of treatment for simple cystitis in the community.

We showed that in women 40 years old and younger, fosfomycin and nitrofurantoin had similar rates of treatment failure (7.96% in both groups, *p*-value = 1); however, patients treated with nitrofurantoin had more cases of reinfection (8.68% vs. 7.47%, *p* value = 0.024). In contrast, in patients older than 40 years, treatment failure was more common in patients treated with fosfomycin (8.31% vs. 6.25% *p* value < 0.001), and reinfection was again more common with nitrofurantoin (9.51% vs. 7.98%, *p* value = 0.0004). When calculating the absolute effect size, it was minor in all subgroups. Practically speaking, the benefit of one antibiotic over the other was small, resulting in a high NNT.

One of the options to define a treatment failure was the prescription of a different antibiotic within seven days. This happened in 1715 patients in our study, 1351 (5.59%) in fosfomycin group and 364 (3.79%) in the nitrofurantoin group. These patients might not be truly failing the first course of treatment, but present longer symptoms which might have caused the physician to consider the option of failure [11]. Intuitively, this happens more often to patients treated with fosfomycin due to the much shorter standard course (one day versus five). Indeed, excluding this criterion of failure, a statistically significant benefit of fosfomycin was observed in all age groups. This is an inherent disadvantage of fosfomycin, and prescribing physicians should take this into account when discussing therapy with the patients and before prescribing a second course of antibiotic within seven days. As noted previously, this effect has only a minor absolute effect size, with a high NNT in all subgroups.

Real world studies have advantages and disadvantages versus randomized and controlled studies. Presumably, patients’ adherence to recommended treatments are much higher in prospective trials, where the investigators can actively encourage the patients to follow the study protocol. Insufficient compliance is supposedly higher with nitrofurantoin due to the need to take the medication several times every day, for five days, compared to the single dose prescription of fosfomycin. However, the number of patients that can be recruited and followed prospectively is limited by financial and logistic barriers, where retrospective studies using a large database retrieved from electronic medical records can include vast numbers of patients. The population included can be much more heterogenic in these studies and be more representative of the relevant population. Another important caveat of retrospective studies is our inability to control many confounders, e.g., sexual activity, menopause status and medical conditions like diabetes and immune deficiency. In a recent study by Martischang et al. [12], risk factors for failure among participants of the Huttner et al. study [9] was only older age and not diabetes or other comorbidities. Decreased renal function was previously described as a risk factor for treatment failure of nitrofurantoin [13], despite not being associated with failure in our study, probably because most of the patients in our study did not have clinically significant renal failure. Microbiological data were not assessed in this study, as was performed in other studies. The main reason for this is that the common practice in the community setting is to avoid taking urine cultures in uncomplicated cystitis, including treatment failures, if there are no symptoms of pyelonephritis. This fact has prevented us from providing data on resistance patterns to either of the investigated antibiotics. Such resistance can of course explain treatment failures; however, in two recent studies from Israel, fosfomycin resistance among urinary pathogens was 2% in military servicewomen [14] and 20.7% to 30.9% among women in Northern Israel [15]. Interestingly, in both studies, resistance to nitrofurantoin was about 30%. This wide range of resistance to fosfomycin demonstrates that resistance is not always associated with treatment failure, as was demonstrated in other studies as well [10,12].

A recent meta-analysis of four studies [16] found no significant difference in clinical or microbiological outcomes between fosfomycin and nitrofurantoin in uncomplicated cystitis, similar to our study. This meta-analysis looked also at the safety of each treatment option and found that the incidence rate of adverse events in both groups was similar.

The limitations of this study are the retrospective nature which did not allow us to establish causality for our results. For example, the endpoint of hospitalization could represent a true failure of the treatment but could also be a coincidental allergic reaction. In addition, we did not have microbiological data as most of the patients in the community setting are treated empirically, as per guidelines [2]. Another important limitation is the lack of data on the actual use of prescribed antibiotics. We did not have access to pharmacy records or other similar sources, and we had to assume that the portion of patients who did not take their prescribed antibiotics was similar between the two study groups. Additionally, it is reasonable to assume that the compliance with the five-day regimen of nitrofurantoin was lower than with the single fosfomycin dose.

In summary, real-world data shows better outcomes of fosfomycin for uncomplicated UTI in patients 40 years of age and younger, with minor absolute effect, whereas in older women, fosfomycin was associated with less reinfections but more treatment failures.

## Figures and Tables

**Figure 1 biomedicines-11-01019-f001:**
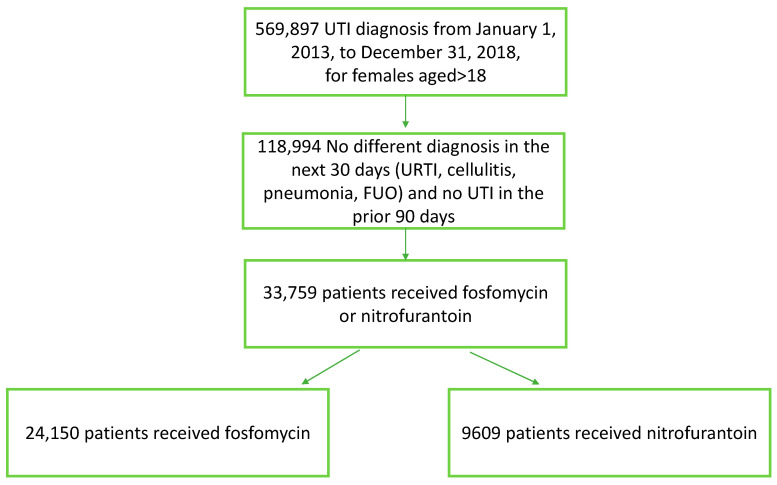
Flow chart of the study cohort selection.

**Table 1 biomedicines-11-01019-t001:** Characteristics of study population; proportions of treatment failure and reinfection.

	Fosfomycin *n* = 24,150	Nitrofurantoin *n* = 9609	*p*-Value	Effect Size	Number Needed to Treat
Age	45.22 (±18.14)	50.62 (±20.05)	<0.0001	0.282	
BMI	26.46 (±7.40)	27.33 (±7.31)	<0.0001	0.118	
Last UTI (in days)	506.32 (±406.50)	407.45 (±380.24)	<0.0001	0.251	
Creatinine	0.76 (±0.26)	0.77 (±0.26)	<0.0001	0.067	
TREATMENT FAILURE (Within 7 days from UTI diagnosis)	1970 (8.16%)	660 (6.87%)	<0.0001	0.049	78
Hospital admission	121 (0.50%)	94 (0.98%)	<0.0001	0.057	210
ER	214 (0.89%)	110 (1.14%)	0.030	0.048	387
Emergency clinic	334 (1.38%)	88 (0.92%)	0.0004	0.043	214
IV antibiotics	19 (0.08%)	13 (0.14%)	0.17	0.018	1766
Change in antibiotic regimen	1351 (5.59%)	364 (3.79%)	<0.0001	0.086	55
Pyelonephritis	66 (0.27%)	73 (0.76%)	<0.0001	0.071	206
TREATMENT FAILURE excluding antibiotic change	690 (2.9%)	334 (3.5%)	0.003	0.034	162
REINFECTION (8th–30th day from UTI diagnosis)	1873 (7.76%)	885 (9.21%)	<0.0001	0.052	69
UTI diagnosis in days 8–30	1173 (4.86%)	592 (6.16%)	<0.0001	0.057	77
Hospital admissions in 30 days	210 (0.87%)	144 (1.50%)	<0.0001	0.059	159
ER in 30 days	361 (1.49%)	155 (1.61%)	0.43	0.010	846
Emergency clinic in 30 days	447 (1.85%)	162 (1.69%)	0.32	0.012	606
COMBINED FAILURE	3843 (15.9%)	1545 (16.1%)	0.72	0.005	604

ER: Emergency room, UTI: urinary tract infection, BMI: body mass index, IV: intravenous.

**Table 2 biomedicines-11-01019-t002:** Odds ratio for different outcomes, in the fosfomycin treatment group versus the nitrofurantoin group, in the multivariate logistic regression model **^#^**.

	Treatment Failure	Reinfection	Combined Failure
Total	OR 0.74 95% CI 0.59–0.92 *p*-value = 0.008	OR = 0.967 (0.8–1.16) *p*-value = 0.71	OR 0.85 (0.733–0.994) *p*-value = 0.4
Less than 40 years	OR = 1.16 (0.7–1.87) *p*-value = 0.56	OR 1.25 (0.794–1.92) *p*-value = 0.327	OR 1.13 (0.786–1.62) *p*-value = 0.49
40 and older	OR = 0.66 (0.51–0.85) *p*-value = 0.001	OR = 0.92 (0.75–1.12) *p*-value = 0.4	OR = 0.8 (0.678–0.948) *p*-value = 0.01

^#^ Controlling for age, creatinine, and time from last UTI diagnosis.

**Table 3 biomedicines-11-01019-t003:** Proportions of treatment failure and reinfection, by age-group (<40 vs. ≥40).

	AGE ≤ 40 Years			AGE > 40 Years		
	Fosfomycin	Nitrofurantoin	*p*-Value	Fosfomycin	Nitrofurantoin	*p*-Value
	*n* = 10,823	*n* = 3469		*n* = 13,327	*n* = 6140	
TREATMENT FAILURE	862 (7.96%)	276 (7.96%)	1	1108 (8.31%)	384 (6.25%)	<0.0001
(Within 7 days from UTI diagnosis)						
Hospital admission	49 (0.45%)	35 (1.01%)	0.0005	94 (0.71%)	48 (0.78%)	0.59
ER	120 (1.11%)	62 (1.79%)	0.003	72 (0.54%)	59 (0.96%)	0.001
Emergency clinic	200 (1.85%)	51 (1.47%)	0.16	134 (1.01%)	37 (0.60%)	0.005
Pyelonephritis	41 (0.38%)	42 (1.21%)	<0.0001	25 (0.19%)	31 (0.50%)	0.0002
IV antibiotics	8 (0.07%)	6 (0.17%)	0.12	11 (0.08%)	7 (0.11%)	0.61
Change in antibiotic regimen	531 (4.91%)	129 (3.72%)	0.003	820 (6.15%)	235 (3.83%)	<0.0001
TREATMENT FAILURE excluding antibiotic change	374 (3.5%)	166 (4.8%)	<0.001	316(2.4%)	168 (2.7)	0.14
REINFECTION	809 (7.47%)	301 (8.68%)	0.024	1064 (7.98%)	584 (9.51%)	0.0004
(8th–30th day from UTI diagnosis)						
UTI diagnosis in days 8–30	437 (4.04%)	159 (4.58%)	0.17	736 (5.52%)	433 (7.05%)	<0.0001
Hospital admissions in 30 days	59 (0.55%)	47 (1.35%)	<0.0001	151 (1.13%)	97 (1.58%)	0.011
ER in 30 days	181 (1.67%)	92 (2.65%)	0.0005	180 (1.35%)	63 (1.03%)	0.061
Emergency clinic in 30 days	275 (2.54%)	92 (2.65%)	0.71	172 (1.29%)	70 (1.14%)	0.4
COMBINED FAILURE	1671 (15.4%)	577(16.6%)	0.098	2172 (16.3%)	968 (15.8%)	0.14

ER: Emergency room, UTI: urinary tract infection, BMI: body mass index, IV: intravenous.

## Data Availability

Due to institutional policy, study data are not published but will be available upon request.

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
