# Peer review of "Real-Life Comparison of Fosfomycin to Nitrofurantoin for the Treatment of Uncomplicated Lower Urinary Tract Infection in Women"

_biomedicines, 2023, doi:10.3390/biomedicines11041019_

Round 1
Reviewer 1 Report
Thank you for your submission.
The data are now old (up to 10 years). With changes in microbial sensitivity and antimicrobial resistance, are the results still of relevance?
Are the patients representative of all women with uncomplicated cystitis? They seem relatively old?
One problem with a very large study like this is that almost every statistical test become significant (e.g. table 1), even when the absolute differences are very small and unlikely to have any clinical significance. Are the very small absolute differences in the main outcomes reported here (treatment failure and reinfection rates) of any meaningful clinical relevance?
There are other key potential confounding variables (e.g. sexual activity, menopause status and conditions like diabetes and immune deficiencies) that were not examined.
Duplication of the same results in tables and figures is unnecessary (delete figures 2-4).
The topic is not novel – there are recent similar studies (examples below) and even a systematic review.
· Sharma S, Verma PK, Rawat V, et al. Fosfomycin versus Nitrofurantoin for the Treatment of Lower UTI in Outpatients. J Lab Physicians 2021;13(2):118-22.
· Konwar M, Gogtay NJ, Ravi R, et al. Evaluation of efficacy and safety of fosfomycin versus nitrofurantoin for the treatment of uncomplicated lower urinary tract infection (UTI) in women - A systematic review and meta-analysis. J Chemother 2022;34(3):139-48.
· Ten Doesschate T, van Haren E, Wijma RA, et al. The effectiveness of nitrofurantoin, fosfomycin and trimethoprim for the treatment of cystitis in relation to renal function. Clin Microbiol Infect 2020;26(10):1355-60.
Author Response
Reviewer 1
Comments and Suggestions for Authors
Thank you for your submission.
The data are now old (up to 10 years). With changes in microbial sensitivity and antimicrobial resistance, are the results still of relevance?
We thank the reviewer for this comment. We believe our data is still very relevant. We added to the discussion some data on resistance patterns in urinary pathogens in Israel.
Are the patients representative of all women with uncomplicated cystitis? They seem relatively old?
Our study, as every other population based real-world study, represent only the participating population. The wide age distribution of our population led us to analyze age sub-groups as well, which indeed has an effect on the outcomes.
One problem with a very large study like this is that almost every statistical test become significant (e.g. table 1), even when the absolute differences are very small and unlikely to have any clinical significance. Are the very small absolute differences in the main outcomes reported here (treatment failure and reinfection rates) of any meaningful clinical relevance?
We thank the reviewer for this comment. To overcome this caveat we calculated the absolute effect size and NNT and added them to our analyses and tables, which enables us to show that the absolute differences between the study groups is not substantial. The aim of our study, however, was to show that fosmomycin is a valid treatment option, despite contrary previous RCT study.
There are other key potential confounding variables (e.g. sexual activity, menopause status and conditions like diabetes and immune deficiencies) that were not examined.
We indeed did not have data on these variables. We added this as a limitation, and a reference to a recent study which did not find an association between any comorbidity to treatment failure in a similar population.
Duplication of the same results in tables and figures is unnecessary (delete figures 2-4).
We have moved these figures to a supplementary file.
The topic is not novel – there are recent similar studies (examples below) and even a systematic review.
- Sharma S, Verma PK, Rawat V, et al. Fosfomycin versus Nitrofurantoin for the Treatment of Lower UTI in Outpatients. J Lab Physicians 2021;13(2):118-22.
- Konwar M, Gogtay NJ, Ravi R, et al. Evaluation of efficacy and safety of fosfomycin versus nitrofurantoin for the treatment of uncomplicated lower urinary tract infection (UTI) in women - A systematic review and meta-analysis. J Chemother 2022;34(3):139-48.
- Ten Doesschate T, van Haren E, Wijma RA, et al. The effectiveness of nitrofurantoin, fosfomycin and trimethoprim for the treatment of cystitis in relation to renal function. Clin Microbiol Infect 2020;26(10):1355-60.
We thank the reviewer for this comment. The mentioned references and others were added to the introduction and discussion sections. Despite not being novel we believe our study of a very large database can add important insights to physicians.
Reviewer 2 Report
I reviewed the article Real-life Comparison of Fosfomycin to Nitrofurantoin for the Treatment of Uncomplicated Lower Urinary Tract Infection in Women, a topic of interest for those who treat urinary infections and for those who are concerned with laboratory diagnosis.
We congratulate the authors for their efforts.
Anyway, there are some aspects that will require improvements
1. r 34-35 please explain grade; for some readers it will be difficult un follow that part of introduction
2.r 46 the purpose is...disect?! It will be better "evaluate"?
3. figures 2,3,4 are difficult to understand; you may change them in other forms
4.the same observation 3 is applied to table 3
5.improve Disscusion
6. add references
7. follow the Instructions for authors
Author Response
Reviewer 2
Comments and Suggestions for Authors
I reviewed the article Real-life Comparison of Fosfomycin to Nitrofurantoin for the Treatment of Uncomplicated Lower Urinary Tract Infection in Women, a topic of interest for those who treat urinary infections and for those who are concerned with laboratory diagnosis.
We congratulate the authors for their efforts.
Anyway, there are some aspects that will require improvements.
- R 34-35 please explain grade; for some readers it will be difficult to follow that part of introduction
We added an explanation for this term.
2.r 46 the purpose is…disect?! It will be better "evaluate"?
Was changed as suggested.
- Figures 2,3,4 are difficult to understand; you may change them in other forms
We have moved these figures to a supplementary file.
4.the same observation 3 is applied to table 3
We have reconstructed table 3 in order to improve its visibility. We believe it is now clearer.
5.improve Discussion
We improved the discussion by comparing our work to others and discussing the different outcomes in our analysis.
- add references
Several relevant references were added.
- follow the Instructions for authors
We have adjusted the manuscript according to the Instructions for Authors.
Reviewer 3 Report
An interesting manuscript comparing the effectiveness of fosfomycin and nitrofurantoin in the treatment of uncomplicated lower urinary tract infections in women is presented.
Considering the significant problem of contemporary medicine, namely antimicrobial resistance and the related need to follow the principles of Antibiotic stewardship, including the application of adequate antibiotics, I consider the article beneficial.
Unfortunately, in its current form, I cannot recommend the manuscript for acceptance and I consider it necessary that the comments below be resolved.
Major comments
1. I recommend supplementing the exact dosing schedules that were used for fosfomycin and nitrofurantoin. Fosfomycin can be used in a single dose, but at the same time it is possible to apply it, for example, in 3 doses, it means 3x3g orally at 3-day intervals.
2. Treatment failure and reinfection can also occur as a result of insufficient compliance with treatment on the part of patients. This possibility is not sufficiently discussed, and although it is mentioned as a limitation of the study, it should be described more detailly. Insufficient compliance is significantly higher with nitrofurantoin and this may explain the higher incidence of reinfection with this antibiotic.
3. I fully understand the fact that the patients were not subjected to a microbiological examination, i.e. determination of the bacterial pathogen and its susceptibility/resistance to antibiotics. However, the resistance of the bacterial agent to fosfomycin or nitrofurantoin is probably the most important cause of treatment failure or reinfection. This fact should be discussed and at the same time additional data should be added in this context. I mean data about the local resistance of the most common bacterial pathogens causing lower urinary tract infections, especially Escherichia coli, to antibiotics. Any differences may also contribute to the discussion when comparing the incidence of treatment failure and reinfection among the antibiotics studied.
Minor comments
1. Non-standard terms for nitrofurantoin (nitrofuranation) are shown in Figure 1.
Author Response
Reviewer 3
An interesting manuscript comparing the effectiveness of fosfomycin and nitrofurantoin in the treatment of uncomplicated lower urinary tract infections in women is presented.
Considering the significant problem of contemporary medicine, namely antimicrobial resistance and the related need to follow the principles of Antibiotic stewardship, including the application of adequate antibiotics, I consider the article beneficial.
Unfortunately, in its current form, I cannot recommend the manuscript for acceptance, and I consider it necessary that the comments below be resolved.
Major comments
- I recommend supplementing the exact dosing schedules that were used for fosfomycin and nitrofurantoin. Fosfomycin can be used in a single dose, but at the same time it is possible to apply it, for example, in 3 doses, it means 3x3g orally at 3-day intervals.
The only schedule available was a single dose. We clarified this in the manuscript.
- Treatment failure and reinfection can also occur as a result of insufficient compliance with treatment on the part of patients. This possibility is not sufficiently discussed, and although it is mentioned as a limitation of the study, it should be described more detailly. Insufficient compliance is significantly higher with nitrofurantoin and this may explain the higher incidence of reinfection with this antibiotic.
We added such an explanation to the discussion, as suggested.
- I fully understand the fact that the patients were not subjected to a microbiological examination, i.e. determination of the bacterial pathogen and its susceptibility/resistance to antibiotics. However, the resistance of the bacterial agent to fosfomycin or nitrofurantoin is probably the most important cause of treatment failure or reinfection. This fact should be discussed and at the same time additional data should be added in this context. I mean data about the local resistance of the most common bacterial pathogens causing lower urinary tract infections, especially Escherichia coli, to antibiotics. Any differences may also contribute to the discussion when comparing the incidence of treatment failure and reinfection among the antibiotics studied.
We added to the discussion relevant data on resistance rates and explained the lack of microbiological data. Additionally, we added data on local resistance.
Minor comments
- Non-standard terms for nitrofurantoin (nitrofuranation) are shown in Figure 1.
We fixed this typo.
Reviewer 4 Report
Dear Editor and Authors,
The article on title “Real-life Comparison of Fosfomycin to Nitrofurantoin for the Treatment of Uncomplicated Lower Urinary Tract Infection in Women” it is quite poorly written.
The introduction is insufficient, citations are few. The topic is interesting, so I decided to review it. However, the manuscript is not at the level of Biomedicines. A manuscript after major writing corrections should be addressed to another journal. However, this does not change the fact that this pledge is scientifically valuable and can be a valuable clue for clinicians, because urinary tract infections account for a large percentage of infections.
Author Response
Reviewer 4
Dear Editor and Authors,
The article on title “Real-life Comparison of Fosfomycin to Nitrofurantoin for the Treatment of Uncomplicated Lower Urinary Tract Infection in Women” it is quite poorly written.
The Introduction is insufficient, citations are few. The topic is interesting, so I decided to review it. However, the manuscript is not at the level of Biomedicines. A manuscript after major writing corrections should be addressed to another journal. However, this does not change the fact that this pledge is scientifically valuable and can be a valuable clue for clinicians, because urinary tract infections account for a large percentage of infections.
We thank the reviewer for these comments. We have revised the manuscript according to the other reviewers' comments and believe it now much improved and suitable for publication at Biomedicines
Round 2
Reviewer 1 Report
Thank you for the revision
Reviewer 3 Report
First of all, I thank the authors of the manuscript for editing the text based on my comments.
The text has been adequately edited and I am now pleased to recommend the manuscript for acceptance.
Reviewer 4 Report
The authors responded to the recommendations and improved the manuscript.